# Exploring the Educational Life Histories of Australian Transgender Faith-Based Secondary School Graduates

Mark Vicars * and Jarrod Wolfe

College of Arts & Education, Victoria University, Melbourne 3011, Australia
* Correspondence: mark.vicars@vu.edu.au

**Abstract:** In this paper we draw on stories of schooling as told by three transgender secondary school graduates. The study does not aim to be generalizable or 'speak' for the educational experiences of all Australian transgender-identifying students. The study is framed by first person articulations of what a trans-positive educational experience might involve. The paper leverages a life-history approach in which the participants rearticulate the influence that cisnormative school environments and media practices had on their transition timeframes. Throughout the life-history interviews conducted in a focus group, the participants considered the concept of how a trans-positive educational approach could be deployed in schools to develop services and resources that align with the findings of the National LGBTIQ Health Alliance, 2020.

**Keywords:** transgender; lived experience; life-history; narrative; transition timeframes; schooling

## 1. Introduction

Educational policies which emphasize inclusivity now require that greater attention be given to the voices which have traditionally been excluded or made invisible. There recently has been a stronger perception in academic circles of a need for inclusivity in education to accommodate those who have been previously categorized into marginality [1].

Research continues to document an increasing reactionary backlash to LGBTIQ2SA+ people globally [2–5] indicating that transgender students experience higher levels of victimization and bullying in secondary schools [6]. Not only are these experienced at higher rates when compared to their cisgender counterparts, but also when compared to their lesbian, gay and bisexual counterparts [7]. Historically victimized, marginalized, and silenced, transgender and gender-diverse discrimination in Australian educational domains continues to have an impact on individuals who do not embody heterogendered teleological narratives. The importance of school policies to support transgender student outcomes, student-led initiatives and activism, peer support and peer intervention in response to in-class instances of transphobia have been identified as salient factors for disrupting misgendering practices. The shift away from the pathologization of transgender individuals as being mentally ill and requiring medical intervention is a relatively recent phenomenon that reflects global changes in attitudes towards understanding and reclassifying transgender as not being a psychiatric illness.

Globally the following cross-national legal regulations have sought to address discriminatory practices with the introduction of the following legislative acts:

2020: Gender Recognition Reform Bill (Scotland) that amends the Gender Recognition Act of 2004, to assist in making the process for individuals to change their legal gender easier.

2017: Adoption of the Yogyakarta Principles on the Application of International Human Rights Law in Relation to Sexual Orientation, Gender Identity, Gender Expression and Sex Characteristics.

2016: Appointment of the first UN Independent Expert on protection again violence and discrimination based on sexual orientation and gender.

2016: United Nations Resolution A/HRC/RES/32/2. Protection against violence and discrimination based on sexual orientation and gender identity.

2015: Sustainable Development Goals. Limited inclusion of people of 'other status' in the non-discrimination section, Paragraph 19 of the outcome document.

2013: United Nations 'Free and Equal' Campaign. A global UN public information campaign aimed at promoting equal rights and fair treatment of LGBTI people.

2013 Diagnostic and Statistical Manual of Mental Disorders (5th ed.) eliminated "gender identity disorder" with "gender dysphoria".

2011: Istanbul Convention supporting protections for gender diversity and LGBTQI+ Individuals.

2006: Yogyakarta Principles. A set of international principles relating to sexual orientation and gender identity; a universal guide to human rights, which affirms binding international legal standards with which all states must comply.

1994: Diagnostic and Statistical Manual of Mental Disorders "transsexualism" was replaced with "gender identity disorder in adults and adolescence".

1980: Diagnostic and Statistical Manual of Mental Disorders (3rd ed) introduced the term "transsexualism".

Socially and culturally there has also been a significant shift in trans visibility in media discourse with initiatives such as Transgender Awareness Week and an annual Transgender Day of Remembrance on November 20 initiated by GLAAD—the world's largest lesbian, gay, bisexual, transgender and queer (LGBTQ) media advocacy organization. Responding to transgender representation in the media domain, GLAAD has become an increasing visible presence in advocating for trans and non-binary affirming representation.

In this paper three trans-identifying participants articulate how in the wake of the Australian same sex marriage plebiscite, transphobia has been utilized in Australia as a more socially acceptable form of bigotry in right-wing political discourse to limit the rights of all LGBTQI2SA+ citizens. They suggest how trans identity can become suppressed as a result of school-based cisgender hegemonies and locate schooling as a place in which the hegemony of conceptual categories of normative sexualities and genders are produced and performed. We understand and appreciate that the identity taxonomies of lesbian, gay, bisexual, transgender, queer/questioning, and intersex are fundamental to the assertion, authoring and authorizing of one's sexual or gendered body and it is important to clarify that the operational definitions utilized in this paper are aligned with the current Sex Discrimination Act 1984. A transgender person is someone who's gender does not exclusively align with the sex assignment at birth and a gender non-conforming person is someone whose personal traits do not match masculine or feminine gender norms. Terms such as 'non-normative', 'non-heterosexual', and 'non-traditional' do not accurately encapsulate a participants embodied sense or experiential expressions of the self. It is important at this point to reference that we have used the participants language of self-definition and representation in their critique of cis–heteronormative discourses to define their own lived experiences [8].

The aim of this paper was not to produce a set of generalizable results, it had no pre-existing theory to test but within an understanding of education as a socio-cultural practice we sought to elicit descriptions of cisgenderism and hetero-centric discourses in faith-based schools utilizing life-history narratives within a focus group. Life-history work offers the re-telling of life stories in a space to the reconstruct narratives of the self to re-write and reconfigure the voice and transform the stories we tell of ourselves and more significantly ourselves in the story. Telling stories can also provide significant opportunities to reflect on the wider socio-cultural narratives which intersect with individual lives; they can become a constructive tool in which the self makes meaning from human experience. The voices drawn upon in this text fluctuate between contradictory cultural identities but are bound by a unity of shared experience and of being subject to a regime of truth. The interpretive locations of the participants were critically examined vis-à-vis exclusionary heteronormative social and cultural practices. If being in the center gives a knowledge of

the center, not of the margins, then it could be said that those who are at the margins have a double knowledge both of the center and of the margins, thus a hidden dialectic exists. Critical dialogue can, we believe, methodologically dislocate normalizing discourses that construct rationales and methods for knowing within webs of power relations.

We propose that that narrative reconstruction of life stories into life histories can reveal the sources and sites of agency. Within their narrative accounts the participants spoke about how the intersections of power shaped by associated religious and political ideologies aligned with, in and by educational institutions. Discussion about binary sex classifications on official documents, gendered dress codes [9,10] and gender-segregated facilities which cannot be avoided by transgender students, inducing fear of transphobic violence in these spaces were raised [11,12] by the participants.

This paper is organized by presenting an overview of the legislative frameworks and policies impacting trans-identifying individuals and is followed by a brief reference of the presence of how colonialism has shaped attitudes towards cisgenderism in the context of Australian life. The methodology, method and analysis drew upon The voice-centered-relational analysis approach prefaces the participant's narratives which are followed by a discussion of the convergences of the lived experiences of the three participant observations and conclusions.

## 2. Legislative Frameworks

The 1990s marked the commencement of the Commonwealth's interest in LGB-TIQ2SA+ issues and in November 1992, Keating's ALP government overturned the Australian Defence Force's ban on gay, lesbian and bisexual people. The passing of the Human Rights (Sexual Conduct) Act in 1994, overruled Tasmania's sodomy laws [13] (the last state to uphold queerphobic laws). The enactment of this law prompted Australian Democrats Senator Sid Spindler to introduce the Sexuality Discrimination Bill in November 1995 to afford protections for transgender people, prior to the existence of state protections. This bill would ultimately fail due to a lack of serious media attention, disagreement on the legal definition of transgender people and right-wing reactionary concerns about participation in sport; and was then reintroduced in 1998 as the Sexuality and Gender Status Discrimination Bill but would not receive support from either major party, thus failing. It was not until 2013 when the Sex Discrimination Act (2013) was passed, which provided anti-discriminatory protections based on sexual orientation, gender identity and intersex status [14]. The failed Sexuality Discrimination Bill and the resultant inquiry of 1996–1997 resulted in increased gay, lesbian and transgender activism and the formation of the Victorian Gay and Lesbian Rights Lobby (VGLRL) in 1997. Due to transgender exclusion from the VGLRL, this led to the formation of the Victorian Transgender Rights Lobby, since renamed to Transgender Victoria in 1999.

After more than a decade of conservative silence on LGBTIQ2SA+ issues, the ALP government in 2008 amended 85 pieces of legislation to provide equal rights to same-sex (de facto) couples and in 2013, they amended the Sex Discrimination Act (2013) to prohibit discrimination on grounds of "sexuality, gender identity or intersex variations". However, the direct use of terms such as 'transgender' were absent due to concerns over inaccuracies of the correct terminology or the potential of causing offence. In 2013, the Australian Federal government passed the Sexual Orientation, Gender Identity and Intersex Status Amendment Act that allows religious schools and organizations to discriminate based on sexual orientation. On 7 December 2017, the Federal Coalition government passed the marriage equality amendment [15]. However, this did not come without religious right transphobic rabblerousing by those who opposed marriage equality and spearheaded the use of transphobic arguments, namely the slippery slope argument, that this legislation would result in gender fluid sex education in schools. In 2017, The Safe School's program which centered around an inclusive curriculum practice was shut down and national debate led to a 2018 review into religious freedom and a subsequent proposal of a Religious Discrimination Bill that was passed in 2019, the Health Legislation

Amendment Bill was also passed in 2019 which did not ban conversion therapy taking place outside the healthcare domain. In 2020 the Education Legislation Amendment (Parental Rights) Bill was proposed to prohibit the teaching of gender fluidity in schools and in 2022 a Brisbane school demanded that families sign anti-gay and anti-trans enrolment contracts which branded homosexuality as "sinful, offensive and destructive" and lumped it into the same category as pedophilia and incest [16].

## 3. Indigenous Conceptions of Gender Diversity and Queer Criminality in Colonial Australia

Prior to colonization, there were several Indigenous Australian languages from central and northern Australia that included words which referred to a third gender. While transgender itself is non-Indigenous terminology [17–19], there are equivalent terminologies used by contemporary gender-diverse Indigenous Australians—sistergirls and brotherboys which hold a relatively long tradition of usage in the Tiwi Islands. However, Australian transgender history has been influenced by the advent of British colonial invasion, influenced by British Law and linked with homosexuality. In 1885, the Criminal Law Act (UK) made homosexual behavior illegal, and all those associated was deemed a capital crime, resulting in harsh punishment from the authorities. This also made people who cross-dressed easy targets for arrest, with many 'cross-dressers' from the late 1800s to the 1930s charged under the Vagrancy Act, as well as other laws against 'offensive behavior'.

The legacy of colonialism is seen in the Australian Commonwealth and State legislation which has been argued to have created systemic institutional discrimination and contributed to the history of mistrust, stigma and the anticipation of abuse in LGBTIQ2SA+ communities in Australia. From the 1970s to the 1990s, transgender people were denied explicit legislative protections from discrimination with Corbett V Corbett, a 1971 British divorce case being the legal precedent cited to deny changes of biological sex and deprive Australian transgender people a legal basis for gender affirmation. Prior to 1996, only the Australian Capital Territory and New South Wales recognized transgender people, who had received gender affirmation surgery (named as transsexuals) as a protected category of people South Australia and the Northern Territory protected "transsexuality [sic]" under their definitions of sexuality and when anti-discrimination laws were passed in 1996, South Australia updated their definition of "transsexuality [sic]" with "chosen gender". Further, amendments to anti-discrimination acts that protect transgender people from discrimination all followed anti-discrimination protections for gay and lesbian people in their respective states as follows: New South Wales in 1996, Western Australia in 2000, Queensland in 2002 and TAS in 2013. However, the Northern Territories still retain their definition and is underrepresented in legislative research in particular in relation to the sistergirl/brotherboy populations of transgender Indigenous Australian peoples. As an outcome of the Labour Government's amendment to the Sex Discrimination Act (2013), state governments have acted in accordance and instituted legislative acknowledgement of gender beyond a binary. The Australian Capital Territory amended the law in 2014 to alter the gender on birth certificates without the provision of surgery. South Australia (2017), Tasmania and Victoria (2019) are considered as the most legislatively progressive, permitting self-identification of gender without further verification from health professionals.

## 4. Methodology of Life History: Working with Narrative Stories from the Field

Dhunpath [20] proposed that the life-history approach is probably the only authentic means of understanding how motives and practices reflect the intimate intersection of institutional and individual experience in the postmodern world. Life history is epistemologically grounded in an interpretive, emic positionality, focused on everyday experiences, and centrally situates the explanations provided by individuals of the world, their reality and actions, which relinquishes researcher ontological control and has the potential to offset the oppressive forces embedded in traditional research methodologies. Life stories are first told by an individual or group of people which then augments itself into life histories after

the addition of further stories, contexts and interpretations that are situated in relation to the original story. Life-history research acknowledges that individual experiences are political and embedded in relationships of power, factors that can often be ignored by more traditional positivist a priori epistemologies [21]. The life-history narratives were composed using verbatim statements from the focus group transcript, and in some cases the participants provided additional email responses that clarified issues raised in the focus group discussion. Each participant was provided with their final narrative for confirmation and further opportunity for comment, to ensure the verisimilitude of their individual voice. The content of the narratives in this paper has not been altered from the transcript, except for the omission of speech disfluency.

As authors we acknowledge that in this paper we are offering a simulacrum of experience [22] and imbricated is an ontologically post-structuralist queer theoretical perspective that seeks to raise the voices of transgender people to determine the type of world in which they wish to live in and to further understand how social justice policies in school settings have fared in the creation of safer spaces for students who identify as transgender or gender non-conforming in Australian School settings.

Within the life-history focus group the participant-led discussions focused on their secondary schooling experiences and the increasing importance of social media platforms for reifying or disrupting gender norms. The role of the researcher within this context was to act as a moderator and guide the conversation, rather than to dictate with excessive questioning and to mitigate against an interviewer-imposed agenda [23]. A brief introduction and context to the study was explained to the participants prior to the virtual focus groups and it was explained to the participants that we were interested in hearing 'ethnographic miniature portraits of their lived experiences'. We provided two research questions to the participants to consider and provide a response for prior to the focus group. These were:

1. Describe an instance of transphobia in the classroom and the merits of peer or teacher intervention, as a good way to handle it.
2. How do we make secondary school a better place for trans students?

## 5. Participants and Research Procedure

Three participants were recruited using purposive sampling, facilitated via a media release to LGBTIQ2SA+ specific services and media outlets [24] including Minus 18, the Victorian Pride Centre, several queer organizations affiliated with Victorian universities and social media websites, such as Twitter, Facebook and Reddit. There were three requirements for participation in this study (i) to be older than 18 years of age, (ii) to have attended a Victorian secondary school and (iii) to self-identify as transgender. A total of nine respondents expressed interested to the preliminary questions but only three respondents affirmatively indicated they would like to participate in the focus group and these individuals were then contacted by email. The selected participants in this study were people over 18 who indicated a willingness to participate in a virtual 35 min focus group. Additional background information about the participants was obtained by email including ethnic and linguistic familial background, if they attended a religious school, their religious beliefs, parental education status, their current educational/employment status and what subjects they enjoyed in school and why. This supplementary information was obtained to contextualize the life-history stories and provide the participants with an opportunity to raise any experiences they might have had of intersectional oppressions [25,26]. Pseudonyms were used for all the participants (see Table 1).

**Table 1.** Participant responses to preliminary questions asked.

| Participant | Ethnic and Linguistic Familial Background | Attended a Religious School | Religious Beliefs | Parents' Educational Status | Participant's Educational Status | Employment Status | Subjects Enjoyed in School |
|---|---|---|---|---|---|---|---|
| Alex Pronouns: they/she | "My usual response to this is that I am Chinese-Vietnamese. Both my parents are Chinese, but they grew up in Vietnam. They emigrated to Australia as teenagers. So there has always been a mix of cultures in my upbringing. We spoke Cantonese at home, but there would occasionally be some Vietnamese conversations or phrases used." | "Yes, my main high school was catholic." | "No religion." | "Both my parents did not complete year 12." | "My highest qualification is an advanced diploma, and I have an unfinished bachelor qualification." | "I am currently employed, it is full time (for the most part)." | "Music, art, and social studies." |
| Naomi Pronouns: she/her | "My usual response to this is that my linguistic background is English and my Ethnic background is broadly European." | "I attended an Anglican school for high school, some of primary and a public school for the remainder." | "I am agnostic." | "My parents have Master's degrees." | "I have a Master's degree." | "I am employed full time as a software engineer." | "My absolute favorite class was woodworking, art and function together was just fantastic. For my final year classes I really enjoyed physics and chemistry, physics because it's just a fascinating topic for me, chemistry because my teacher was great." |
| Kristen Pronouns: she/her | "My usual response to this is that my I am a second-generation migrant of Chinese descent, born and grew up in Melbourne. I speak both Shanghainese and Mandarin with my extended family." | "I did attend a Presbyterian religious school." | "Atheist." | "Bachelor's Degree." | "High school/some university (did not graduate)." | "Freelance music journalist." | "Most enjoyed music and English, because I got to relate to, and express myself through art-which has been an important theme in my life since." |

*5.1. Summary of Participants–Background Information*

Once the focus group discussion was completed, to ensure interpretative saturation we leveraged a voice-centered-relational method to explore the variety of participant perspectives, their interactions in relationships and to reconstruct their voices as coherent narratives [24].

*5.2. Data Analysis Strategy: Voice-Centered-Relational Method*

Following participant verification, four readings of the transcript in accordance with voice-centered relational method (VCRM) were completed. The voice-centered-relational method (VCRM) is a feminist methodology originally created within a psychological discourse. To obtain interpretive saturation, four distinct readings of the data were obtained. Multiple readings, following participant verification, in accordance with VCRM were completed (i) considering one's own personal relationship to the speaker, including one's own biases; (ii), reading for the voice of 'I', focusing on the respondents' experiences, feelings and speaking about themselves as a subject; (iii), reading for relationships and (iv), placing people within cultural and social structures. The narratives of each participant were then drafted and sent back to each participant to confirm, add comments and amend any misinterpretations. The first reading established many of the themes that are later discussed in this paper. For the second reading, I-poems was adopted, with all "I", "me" and "my" statements included in the reading and analysis allowing the researchers to identify the participants' unique voice and create a space for them to speak of themselves in their own body before they are spoken of. The second reading was particularly important due to the emphasis on the existential realities of transgender people and their subjectivities as such forming the basis of the narratives. For the third reading, specific stories were conveyed via vague representations of mostly informal, but organized bodies within the school, including school administrations, teaching staff and other students. Further, vague representations of the 'other' were explored in this discussion; this was identified as the wider Australian public and their externalized perceptions of transgender people, contained within online and media discourses. The fourth reading established three social and political contexts, as informed by their schooling and present life experiences, influenced by media and political discourses that existed, shaped and continues to shape their lives, expression and thought. It was not until this final reading that a clear chronology was established and the redrafted narratives of each participant were sent again to each of the participants to confirm, add comments and amend any misinterpretations. As authors we acknowledge that in this paper we are offering a simulacrum of the data through our reconstruction of their stories [25,27].

You got to be seen to be heard: Graduate Stories.

5.2.1. Alex's Story

"I've only recently been transitioning. I went to a religious school until year 11, doing it at a Catholic all-boys school because I wanted to be with my friends. I felt completely alienated from high school, there was not much talk about transness at the time, I do not even remember having much sexual education, it was kind of like brushed past, which makes sense because of the religious and conservative nature of the school. I also remember it was popular to be like crude and reference Nazis and racism, the cruder you were, the funnier it was. I would often see bullying in the form of gender policing, calling others 'girls' or 'gay.' Gender policing was also committed by the teachers at my first school, I had long hair at the time and there would be comments about me needing a haircut or jokingly asking if I could see from behind my fringe. I was absent a lot from that school, skipping a lot of days and not doing so well. Because there was talk about different sexualities; and just feeling a bit different, I was like, 'oh maybe, maybe I am gay, maybe that is it'. However, really, that was not really it, not completely anyway; I always just thought that I was gay, or bisexual. I did year 12 somewhere else and there was an underlying sense of acceptance. There was no uniform and students generally presented as they wanted. While

gender policing occurred at my second school, again, calling other's 'girls' or 'gay', I still felt that sense of acceptance. I even saw a male student dress in women's clothing for a school fashion show. I had never seen that before in people of my age group and was a little shocked at the time. Now that I think about it, I am glad I was in an environment where a student could not only challenge gender norms, but have it embraced and presented to an audience. It's scary that certain attitudes, in this case, attitudes towards trans people, can become unlocalized. With the internet; the information and attitudes can spread and be manipulated. The fear that I have is some of the US rhetoric making its way to Australia; if you look at the COVID conspiracies, that probably made its way here through like social media and Facebook memes. Remembering high school when it was popular to be crude on the internet, while that still goes on, I feel like it's different. I just hope that when kids see it, they're not taking it seriously and becoming radicalized. Which I do not think that they do, they might understand the irony a bit more than some people give them credit for. However, it is nowhere near as bad as the US and I am optimistic that things have gotten better and will continue to get better here. I think peer intervention will always be stronger, it shows support from the same level of power and a lot of bullying comes from wanting to be cool; the social pressure is a lot stronger than the slap on the wrist. I do find making the bathrooms all gender neutral would help to reduce that binary thought, giving students more freedom to express and explore what their identity is. Maybe removing uniforms all together, you would not just go for like either the boy's uniform or the girl's uniform, break way the gendered part of it. It would just be: do you want a shirt? Do you want a dress? Just trying to identify where that binary comes in as much as possible, as the organizers of a school and then, figuring out ways to reduce that informed by professionals".

### 5.2.2. Kristen's Story

"I went to a Presbyterian school on a scholarship from 2003 to 2008, and it was drilled into me that it was all about academic achievement, when obviously a teenager's development is about so much more than that. Being so caught up in the rituals and the church and singing of hymns, it felt like I was not allowed to develop as an individual and it stuck in there for so long that I did not really learn to fully unpack it until last year, which is when I started transitioning. I feel an institution that is inherently religious, conservative and built on the gender binary is never really going to be able to fully support a trans student, no matter how much lip service they pay. If I had been out and trans during high school, like who knows how it would have gone; overall, it was a dead end for me. I think there was a recent story, about a trans girl in year 12 come out and I think her decision was to like to stay at the school, because she felt quite supported by her year group and potentially by the faculty as well. I do not know if that is the best-case scenario for a trans student at a single sex, boy's school, it seems like there's more to it than that. The potential for transphobia is always going to be there until broader society really learns to deconstruct and deal with it. I feel like so much media commentary about us is gatekeepy at best and concern trolling at worst. I hope that is something that changes soon and that there is more like room for trans voices within mainstream media; not just for us but for all the kids who are in high school who might be questioning and looking for answers, they need role models as well. However, I do get a broad sense that culturally, gen-z is dealing with this stuff better and is more aware of issues around race, gender, sexuality and intersectionality, though it's hard to gauge how progressive a generation is. It is something I could not even dream of when I went to high school, the concept of trans and access to knowledge online and media representation was barely existent. In general, I feel optimistic about trans rights in Australia, I was happy with the response to the religious school's bill in February, which was a very swift backlash. However, the fact that it was trying to be passed at all is very bad. Though I feel pretty good about the direction that we are going in and I do not see the Murdoch media for example, being able to weaponize transphobia to their benefit, in the same way that they have in the UK and the US That does not mean that like everything is perfect, but I feel ok and relatively safe and supported here. Certainly, things in the US are

very worrying, but I am not sure like what more I can say or add to that conversation. It would be good to have incorporated curriculum within Safe Schools that talks about trans rights and trans history, just because we're so often portrayed as a recent phenomenon and that is not true at all. The deliberate erasure of that is inherently harmful. It is something that needs to be developed in contingency with trans people and trans educators, it cannot just be left to allyship. I feel like transness is something that is very much a cultural dialogue and in flux-there is not a single, correct way to be trans. This makes it something that is difficult to teach, because it requires dismantling the gender binary almost entirely and cannot be tackled in a single class. Transness is more than pronouns, which is so often like how its portrayed to be. I think it's important for us to move past the idea that there is a 'trans debate', while dismantling some of the more noxious recurring tropes that pop up, like the constant media fixation on bathrooms, detransitioners, puberty blockers, etc. Although it is worth talking about them with sensitivity, those discourses are given far more weight than they have in trans people's day-to-day existence".

5.2.3. Naomi's Story

"I guess technically I went to a religious school, they were Anglican, but we did not really practice anything, I think we went to church maybe one time. It was a boy's school that then merged with an all-girls school. We had separated classes for the lower year levels, but the higher year levels were smooshed together. I was actually kind of happy about that, when people ask me where I went to school for high school, I can say, 'yeah I went to this girl's school'; they're like, 'oh, cool, makes sense. 'Even though I did end up at an all-girls school, the first time I really thought about it, I was like, this is going to be too difficult, I cannot stay here because I do not feel safe. However, I also do not know if I would feel safer elsewhere. There were definitely signs and one day it hits you like a truck, the door was opened and then sun shone again; and while I had an inkling, I was not ready to come out. School would have been a bit different had those feelings been processed, but there was not that sort of environment to think about those sorts of things. At least my experience with sex-ed classes, we barely even talked about people being gay. Growing up in the era of 'Haha, you're gay' it is pretty difficult to navigate any of this when no one's talking about it in a way that is not pointing a finger and going 'Ha, would not want to be that'. There needs to be some sort of support in place, it would be nice to have a generalized approach to sex-ed that includes gender diversity. I remember the first time I really stared questioning, around the time Caitlyn Jenner came out, the media was just a massive dumpster fire; they were just like, 'he this' and 'he that' and 'his Olympic medals. 'I feel like we have moved mostly past that, like people do not deliberately make that choice to misgender people anymore. It is heading in the right direction, I know some people who have kids, some of them have been questioning, some of them have come out as asexual and everyone that they talk to in that space is kind of, is very aware of it, very accepting of it. When there are headlines in the news that try to make trans people or gender diverse people look bad, the parents are outraged, they're just like, 'this is unacceptable, just because my kid is not, does not mean everyone should not be safe'. So, I feel like we could end up in a place soon where the media stops portraying us as the bad guys, which would be nice. Of all the places to be trans, Australia is a pretty good place. Though, here is a list of things that do not concern me about the US at the moment. A very long list. In the UK at least there is outrage, there are many people that are very upset about how trans people are treated and there is some progress being made over there. I think with the internet being such a big place, yes there is the expression of all the right-wing people that are just dumpster fires; but, at the same extension, there is a lot more information for people to see about gender diverse people. Having the information makes them a rounded, exposed and less inclined to just look at one thing and then latch on to that as the singular idea. I guess it's more up to the parents at that point to have raised a kid that is looking at different places to get different information rather than just listening to a single person. I think that is what gives me hope is that most kids that, you know, are not in terrible situations, they do have a brain

in their head, they do make sense of things. I think kids are smarter than most people give them credit for".

### 6. Discussion—To Be Seen or Not to Be Seen

All three participants utilized feminine pronouns and had attended faith-based all-boys schools for at least five out of the six years of secondary schooling. They referenced to varying degrees how religious ideology had a permeating influence on their respective contexts resulting in their transitions being delayed until later in their adult life. A chronological assessment of the participants' stories indicate an increase in trans-visibility over time from transphobic silence in the mid-2000s to the present day. The media and political representation, as narrated by our participants' perceptions, expressed how an understanding of "transness" has been due to the extent of media and political discourse which has impacted on their lived experiences. Alex, who attended secondary school from 2002 to 2007 closely aligned their experience with Kristen's experience from 2003 to 2008, in the sense that "there was not much talk about transness at the time that I was in high school". Moving to the subsequent decade, Naomi's experience introduced the emergence of transphobic media discourse whilst maintaining an absence of formal transgender educative discourse within a cisnormative regime. This timeline of emerging transphobia where explicit homophobic language transitioned to an emergence of transphobic language timeline aligns with US Survey data showing a decrease in homophobia experienced by students from 2015 to 2017, but an increase in transphobia between 2013 and 2015 [27]. While some might argue that homophobia in Australia is becoming considered as a completely inappropriate tool for political discourse, transphobia is being increasingly used to marginalize and exclude as most recently observed in debates in relation to transgender people in sport [28].

The participants noted how the relatively higher presence of religious ideology impacted on their transition timeframes and several studies have shown the connection between religiosity and transphobia [29–35], which is particularly heightened and aligned with conservative social values, the reliance on tradition, and the predictability and alliance with right-wing political forces. Kristen, who attended a Presbyterian school, spoke of the highest involvement of religious influence in school cultural practices and did not transition for 14 years until after graduating from their secondary school. Alex, who went to an all-boys Catholic school, received minimal sexual education and thusly attributed their alienation exclusively to their sexual diversity, experiencing a delayed transition until later adulthood. In contrast, Naomi's school experience commenced in an Anglican all-boys school that later merged with an all-girls school, becoming a co-educational school in senior years. She described the limited influence of religious rituals and ideology throughout her schooling and contemplated transitioning during secondary school and transitioned relatively soon after her secondary schooling graduation in comparison to the other two participants.

Convergences between the decades of the participants contemplating living openly as transgender during secondary school was brought up in the context of a news story concerning a transgender student not being supported by and in the educational domain. All three participants in response to acts of classroom transphobia expressed a peer-led response would be more effective to mitigate bullying and a verbal act transphobia compared to a teacher-centric response. This portion of the discussion acknowledged the limits of classroom management with a disciplinary response from peers being preferred.

The participants' justifications were made on grounds that peers were perceived as having a collective social power that could impose relatable ethical standards and appropriate justice [36–38] while the teacher, acts as a singular authority figure of non-equivalent power and is separate to that of the students. Alex remarked "I think peer intervention will always be stronger . . . it shows support from like the same level". Kristen posited that a teacher and the wider schooling apparatus may be required to step in when

the danger of escalation is present as the risks of transphobic violence are higher than those of cisgender students [39–44].

Due to the framing of the two questions asked, discussion was limited to in-class acts of transphobic violence; acts that occurred outside the classroom were not discussed. Sexualized gender violence was noted to typically occur outside the auspices of teacher observations and primarily in the toilets and in undefined areas of the school, associated with limited teacher observation, such as hallways, recreational areas [45] or outside the school walls.

The participants' perspectives support existing research outlining the importance of peer intervention in response to transphobia and emphasizes how such a behavioral model that situates student-centered action as a demonstration of social justice allyship can assist teachers to overcome their own anxieties about their perceived transgender knowledge gaps and resultant inaction to these instances. To develop this mechanism further the participants suggested teachers, transgender students and the school administration could work collaboratively to develop policies and procedures to supports the development of classroom behavioral management plans that permit students to respond to transphobic acts of speech and support transgender peers.

Whilst the participants maintained an optimistic disposition regarding the progress of transgender rights in Australia, anxieties of an existential threat in the form of foreign media discourse and its ability to incite transphobic violence was raised in the discussion. Kirsten commented "I feel pretty optimistic about trans rights in Australia in general; the response to the Religious Schools Bill in Feb, was like very, a very swift backlash and, I am happy with how the response went but the fact that it was trying to be passed at all is very bad".

## 7. The Rhetoric of Australian and Anglophone Media Discourse

While 'transgender' is a relatively new term in Australian media discourse, research stories about 'gender crossing' have been existent in Australian newspapers from the 1870s. While not all these covered subjects were transgender individuals, the media's interest in individuals that challenge gender normativity does have a 150-year history in Australia [46]. In the 1950s, the Australian tabloid The Truth, in addition to support from psychiatric discourse, was instrumental in the propagation of hysteria and the framing of criminality surrounding transgender women [46].

The term 'transgender' has existed in Australian media discourse since at least 1965, but this did not become common vernacular until the 1990s, from the 1970s inappropriate terms, such as 'transvestite' and 'transexual', were widely used interchangeably to describe transgender individuals [47–49] and was instrumental in forging self-identities through continued recognition from others, as recognized members of the same category as oneself. Further, it influenced the articulation of policy that has impacted upon the lives of transgender people. Forty years later sensationalistic discourse in the 1990s inspired by the British Media positioned the language of 'sex change' as a foundation of deception prominently featured in hate crimes in the same decade. The legislative legacy continues to uphold a contemporaneous, pejorative nomenclature 'trap' that is increasingly utilized in online media discourses and associated with legacies of transphobic violence.

The participants suggested contemporary online media platforms frequently utilizing transphobia as a mechanism of gender policing [50–52] and the ease in which adolescents can encounter transphobic discourse organized around the nexus of online conservative groups is of serious concern in virtual spaces. In social media contexts, the participants noted how discussions of issues concerning transgender people often result in debate and the subsequent delegitimization of transgender people. Discussion topics such as gender-neutral bathrooms often construct narratives of a zero-sum game, framing the debate around lost privileges and declarations of victimhood. This tactic of 'digressive victimhood', whereby the adoption of victimization status by members of dominant groups

is utilized to mitigate accusations of discrimination and subsequent threat-minimization is potentially harmful for positive trans-visibility.

As social justice and inclusion carry with them a conceptually multi-faceted set of coded meanings requiring constant review, reconsideration and re/textualization, social network sites are capable of agitating normative discourses and exclusionary practices. Herrara proclaimed:

"Social media has highlighted the significance of social relationships and it has been noted how social media platforms make money off our hearts racing . . . Even if we get thousands of supportive messages, our brains will focus on negative interactions, tricking us into a fight or flight response. Before we know it, we are arguing with a person who may not even exist" [53].

Of primary concern noted by the participants was the Intellectual Dark Web (IDW), a group of politically conservative public intellectuals, podcasters and political commentators, many of whom, espouse transphobic perspectives. These notable IDW figures possess large platforms on YouTube, with each channel receiving between 25 million and 1 billion views and are potentially accessible by 81% of internet users aged between 15 and 25 that access YouTube, the second most-visited website on the internet. Research into the political bias associated with YouTube's algorithm demonstrated the potential for YouTube users to be pointed towards alt-right content and radicalized as YouTube's algorithm "frequently suggest(s)" alt-right and IDW content to users. While the potential for young YouTube users to be radicalized is alarming, the participants noted how the IDW and platforms express transphobic perspectives and deny the lived experience of transgender youth primarily through their utilization of tactics such as explicitly misgendering transgender people. As the IDW maintains an extensive presence on social media websites, the intersection of technology and education unites various discourses as a mechanism to maintain cisgender hegemony. They articulated a concern for the incorporation of YouTube for valid educative purposes in Australian secondary school contexts, permitting access to transphobic content from authoritative sources. Without adequate resistance to support transgender students, these online platforms produce and disseminate transphobic, cisnormative vernacular and rhetoric. It has been suggested how such social media platforms can perpetuate existing transphobic perspectives with continually biased content. Ever mindful to ask questions concerning how identity, power and social beings are organized in educational domains, we suggest the processes and cultures that keep cisnormativity in operation are tacitly understood in schooling contexts and routinely socially and politically conceptualized in terms of difficult knowledge by adults. Articulating a counter-narrative that interrupts the silencing is connected to reconstructing the problematic and contestable character of knowledge claims that include:

- the presence of underlying values and assumptions (including those embodied in competing paradigms, discourses, rationalities and ethical principles);
- recognition that 'meanings' can be conveyed through communicative devices other than formally reasoned argument, e.g., via sound and visual images, by the use of rhetoric;
- the significance of knowledge communities and cultures, and of claim-makers' interests and allegiances;
- the associations between knowledge and power.

## 8. Conclusions

As the official curriculum struggles with representing diverse cultural and social positions it is important to counter a deficit mindset and highlight the potential for trans-liberatory resistance via speech acts. Social media can provide a space for the articulation of inclusion affording opportunities to question taken-for-granted assumptions, beliefs and values and interrupt performative narratives of cis-normalcy. From the observed period, 2002 to the present day, there is a progression from transphobic silencing to public exposure and the attainment of social progression which unfortunately can bring a reactionary back-

lash. However, with each public transgender story transgender discourse has developed in conjunction with an emerging acknowledgement of a trans-liberatory and inclusive future. All three participants attended all-boys schools for a significant portion of time and these contexts, as identified by Kristen, are founded on a teleology that mandates different social roles, reinforcing binarism and making men and women into exclusive, fixed and natural categories. Discussion concerning binarism arose and the participants articulated how commonplace it can be to observe a climate of tension due to the presence of reactionary forces who cling tightly to hegemony and perceive divergence from normative behavior as a threat.

Focus group discussion regarding the importance of the explicit inclusion of transgender and gender-diverse learning materials occurred in response to a direct question, "how we can make secondary school a better place for transgender students?" The participants offered several recommendations. Firstly, Kristen offered that an incorporated curriculum that explicitly taught gender diversity and transgender histories, with support from Safe Schools, would help combat erasure and invisibility. Secondly, Naomi offered the importance of literature that includes gender diversity as a normalized background.

McKinnon [5] in their chapter on 'Efforts to Support LGBTQ Children in Australian Schools in the 1980s and 2000s' makes a statement about the perpetuation of conservative heteronormative mythos by the strategic use of paternalism, embodied in the idea of the "figurative child (which) continues to symbolise Australia's ideal future, rich with potential, but only as a white, straight and cisgender citizen" (p. 736). Naomi, the youngest participant, advised that there was barely any teaching of diverse sexualities and her timeframe aligns with a 2014 report on Australian transgender and gender-diverse students [54] in which 66% of respondents indicated transgender and gender-diverse sexuality education was inappropriate. This figure was higher for students at Catholic or other Christian schools, with a rate of 85% saying it was inappropriate. The framing of sexual education in the majority of Judeo-Christian English-speaking countries maintain a focus on cisgender heterosexuality with limited discussion about sexual pleasure, or positive depictions of body except for cisgender men. Once cisgendered normativity becomes institutionalized, the processes and cultures that keep it in operation are tacitly understood by those who inhabit and reify the norm, further shaped by an education system which is in turn is impacted by legal and legislative frameworks that provide a foundational ground of operations. Media discourses which operate somewhat independently and within their own intersections of power from a multitude of political agendas and associated discourses have the potential for the provision of disseminating transgender ontological potentialities. These factors all influence gender transition timeframes. This can be seen in the contrast between Naomi's story of increased exposure to transphobic media reporting in 2015 and her emergent transgender awareness and Kristen who transitioned much later in life.

Despite the optimism held by the participants for a trans-liberatory and inclusive future, a climate of tension persists, due to the presence of reactionary forces who cling tightly to hegemony and perceive divergence from normative behavior as a threat. The importance of policies that support flexible gender expression and the removal of superfluous gendered school practices has been articulated as important in the lived experiences of the participants. How these forces intersect within educational institutions and continue to be shaped by associated religious and political ideologies and their aligned authoritative bodies remains a troubling presence. The presence of explicit transphobic discourse can force a retaliation of transgender youth in response to explicit transphobia and assert themselves through a position of 'vulnerability-in-resistance' Such a parallel can be drawn by teachers in the UK who, in response to the passing of the explicitly homophobic Section 28 in 1988, were able to speak out against the explicit silencing intended by that legislation. In short, solidarity is key, as it always is to achieve social justice and to mitigate the silencing surrounding trans issues in education. The participants agreed that any consideration of a transgender-related curriculum needs to be developed "in contingency with trans people and trans educators". Kristen expressed her dissatisfaction with the limitations of ally-

ship, which was felt by all participants and all the participants outlined that the diversity of transgender perspectives and experiences can be best expressed by trans-identifying individuals and are best positioned to inform cisnormative contexts and disrupt a singular conception of transgender ontology.

The limitations of the study started from the onset of participant selection, with the process taking a relatively long period of time to recruit a small sample of participants. The complexities around the self-disclosure of identifying as transgender or gender non-conforming amongst young people was raised to the researchers. Alex, Kristen and Naomi suggested that research conducted by a transgender researcher could have garnered a larger volume of participants via increased access to the transgender community and more could have been achieved if the researchers had made use of 'insider' subcultural capital. Extending the overall time of the focus group and the inclusion of additional questions to explore lived experience would have benefitted the study. As a starting point for future research this paper has involved consideration of how transgender discourse is situated and constructed on a national level but is enacted in different legislative state frameworks. The inclusion of participants from other Australian states would have aided the identification of convergences or dissonances within each state education system. This information would have afforded a more detailed and specific understanding of legislative changes and their resultant influence on curriculum regimes in educational domains. This information could then inform an understanding of the impact on transgender ontologies and transitioning timeframes.

**Author Contributions:** M.V. 60% and J.W. 40%. All authors have read and agreed to the published version of the manuscript.

**Funding:** This research received no external funding.

**Institutional Review Board Statement:** The study was conducted in accordance with the Declaration of Helsinki, and approved by the Institutional Ethics Committee of Victoria University HRE 16-295 on 5 December 2022 for studies involving humans.

**Informed Consent Statement:** Informed consent was obtained from all subjects involved in the study.

**Data Availability Statement:** Not applicable.

**Conflicts of Interest:** The authors declare no conflict of interest.

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
