# Peer review of "Exploring the Educational Life Histories of Australian Transgender Faith-Based Secondary School Graduates"

_2673-995X, doi:10.3390/youth3010015_

Round 1

Reviewer 1 Report

The article is interesting and, overall, I think it is well written and significant. I particularly liked the reference to intersectional inequalities/privileges of trans people according to their social positions (in the discussion). 

However, there are a few issues that I think would help to improve the structure and arguments of the article:

·      Introduction:

o   I believe that the authors should state their main arguments in the end of the introduction; followed by a brief description of the structure of the article;

o   I would prefer to see the definitions/discussion of key concepts, such as sex, gender, trans, gender diversity, sexual orientation, gender performance… in the introduction and/or when they are first mentioned;

·      Background:

o   I would like to have a bit more information, in this section, about trans and gender diverse youth in general, in Australia, and at (high)schools in particular. The authors could also mention the importance of LGBT+ and trans activism;

o   The authors mention the Commonwealth in relation to Australian trans and gender diversity related laws and the influence of British media. The authors could very briefly mention Australia’s position in a (post)colonial world;

o   I think that the attention that the authors pay to the internet and to transphobic discourses is very import. However, I believe that the authors could summarize that section to include the issues that I have mentioned above. Importantly, the authors could also mention the importance that the internet has in transmitting knowledge and creating networks for trans people;

·      Methodology:

o   My understanding is that the authors are using qualitative methodological approaches; and that the method of data collection is a focus group;

o   Can the authors clarify why they had more people responding to their initial contact, but only had three people participating in the focus group? For me it was not completely clear if these participants were the only ones willing to participate in the focus group or if there was any other reason;

o   Can the authors say how long did the focus group take and where did it take place?

o   I would put the characterization of the participants in the methodological section;

·      Results:

o   I would start this section with the significance of telling trans stories, particularly in relation to their research question.

Author Response

I believe that the authors should state their main arguments in the end of the introduction; followed by a brief description of the structure of the article;

A:  Main idea introduced in the introduction of the article and structure made explicit

o   I would prefer to see the definitions/discussion of key concepts, such as sex, gender, trans, gender diversity, sexual orientation, gender performance… in the introduction and/or when they are first mentioned;

A: These have been provided in the introduction

  • Background:

o   I would like to have a bit more information, in this section, about trans and gender diverse youth in general, in Australia, and at (high)schools in particular. The authors could also mention the importance of LGBT+ and trans activism;

o   The authors mention the Commonwealth in relation to Australian trans and gender diversity related laws and the influence of British media. The authors could very briefly mention Australia’s position in a (post)colonial world;

o   I think that the attention that the authors pay to the internet and to transphobic discourses is very import. However, I believe that the authors could summarize that section to include the issues that I have mentioned above. Importantly, the authors could also mention the importance that the internet has in transmitting knowledge and creating networks for trans people;

  • Methodology:

o   My understanding is that the authors are using qualitative methodological approaches; and that the method of data collection is a focus group;

o   Can the authors clarify why they had more people responding to their initial contact, but only had three people participating in the focus group? For me it was not completely clear if these participants were the only ones willing to participate in the focus group or if there was any other reason;

o   Can the authors say how long did the focus group take and where did it take place?

o   I would put the characterization of the participants in the methodological section;

  • Results:

o   I would start this section with the significance of telling trans stories, particularly in relation to their research question.

Reviewer 2 Report

i recomended this artcle without change

Author Response

No revisions required in response to reviewer 2 assessment of the paper. English language and spell check completed as advised.

Reviewer 3 Report

All the comments on the article are in the attachment.

I did not comment on style and formatting in the review, but there are a lot of format mistakes (e.g. double spacing, missing dots and commas).

Author Response

REVIEW
Journal: Youth (ISSN 2673-995X)
Manuscript ID: youth-1937298
Title: Exploring the Barriers to Enhancing Educational Outcomes of Transgender
Students in Australia

Dear Reviewer,

Thank you for your detailed suggestions to improve the paper- they are most appreciated and welcomed.  We have addressed all your suggestions for improving the paper and these are outlined and highlighted below:

I believe that the authors should state their main arguments in the end of the introduction; followed by a brief description of the structure of the article;

This has been included as suggested

Title and abstract                               
The title and the abstract are not fully adjusted to the general aim and content of the article. Title is not fully pointing out the key issues in the title the term “Exploring the Barriers to Enhancing Educational Outcomes” -it is not fully in line with the content of the article. The article is not exploring barriers to enhancing educational outcomes. It rather explores the possible ways of enhancing educational practice. The “educational outcomes” usually are about educational content and aims of education (teaching process, materials ect, todevelop knowledge, skills ect. I do not exactly understand how the authors understand that term “educational outcomes”, as within article they focus rather on educational environment than on “educational outcomes”. Authors within article use term “transgender educative outcomes”. What does it mean – the transgender students should achieve different outcomes than cis students? Or do the author mean the educative outcomes applying for all the students (cis and trans), but with introducing outcomes about transgender man?

This have been amended  and  clarified as suggested

- the authors write about “transgender students in Australia”, but they based article on 3 participants from religious schools, so the title is misleading (should be somehow limited – if not the title, than at least abstract should clearly mention it;

This have been amended  and  clarified as suggested

- abstract should contain some more detailed information about method used in the qualitative approach. The abstract brings mix of participants statements, authors interpretations, other author research results and some contextual categories. This mix makes it extremely difficult to understand which elements from abstract mirror
the transgender students perception and experiences, and which are authors’ (and other researchers’) interpretations. This is not typical to use so many references within abstract.

This have been amended  and  clarified as suggested

Keywords:
- disciplining normative gaze – this term is not being used within article, so it is not relevant to place it as a keyword
- reflexive story-telling – this term is also not being used – authors write about telling the stories, but in methodology they do not describe reflexive telling as the approach or strategy for research, so it is not relevant to use it as a keyword

This have been amended  and  clarified as suggested

Introduction
- there are several very long sentences, which are vogue and difficult to fully
understand authors thoughts, as well as the way of references – it is not clear to
which part of the sentence those references belong (e.g. lines 39-44; 44-50). I suggest
to simplify the style to make it more clear and readable,

- Line 39-44 – authors confession about being cis-gender males within this particular
sentence is not clear. I do not exactly understand the reason for giving it – is it like a
justification for the research or way to show objectiveness of the observations (as not
being trans man). The sentence also takes some insight into the reason to start the
research in the field.

This have been removed

- line 44-52 – is there a quotation (lines 47-52)? – it’s not clear, where the sentence
finishes and whose quotation is being used? Author show in this part media usage
and it’s influence on shaping hetero hegemony. They omit the reversed effect (media,
popular culture ect.) as a source for sharing experiences, making trans people visible
– I suggest to add this positive effect as well- line 48 – glocal discourses – shouldn’t it be global? (I know term “glocal”, but it
doesn’t fully corresponds with meaning of the sentence)- lines 53-55 – ambiguous sentence – what does it mean “identifying graduates via a
focus group” ?? Is it about identifying their experiences??- line 56 “improve transgender educative outcomes for transgender students” – what
does it mean “transgender educative outcomes” – can educative outcomes be
transgender?- line 60 – where is the end of quotation (is it the research question)??- legislative frameworks and Visibility - are those elements of Introduction – if so,
they should be somehow marked, numbered to make it clear;- legislative frameworks – in this section there is a missing element about situation of
transgender students at schools (their rights ect). As the article focus on experiences
from schooling, there should be more focus on children and adolescents, e.g. rights
and process of recognition of self-identification of gender or short description ofprocedure taken in cases of youngsters.

3
- visibility - the focus is taken on negative media impact. Although authors signalize
the positive outcomes, they are limited to short mentions, I believe it should be morestressed (positive media impact on transgender), e.g. there are given names of IDWrepresentatives – there might be also given counterexamples of people supporting
LGBT+ (e.g. Boy George ect.)
- line 128-132 – I do not understand the narrative and this particular paragraph (is itquotation?), this paragraph seems to be not coherent with previous and next one
The missing elements of the introduction:
- there could be clear given clear definition of transgender/ cis gender people ect. (itmight be consider as obvious for authors, but clear defining basic terms is necessary,
- authors focus strongly on Australian context, which is well described (with some
elements from other countries experience, e.g. British). There should be at least
paragraph in introduction about the global changes in perception, legislation overthe topic of transgender man (e.g. Istanbul Convention, changes in DSM
classifications should be mentioned, with showing process of depenalization andoverall switch into more inclusive global trends);
- authors should deliver some theoretical background over the perspective of
analysis. It’s about overall theoretical background and also about the detailed
elements. E.g. for legislation elements and media M. Foucault concept of power anddiscourse analysis should be taken into account, about visibility the concept ofsymbolic reality and socially constructed reality (e.g. v. Dijk, Giddens might be takennto account). For the overall categories –the pedagogy of oppressed (Freire), stigma(Goffman) should be mentioned. The theoretical categories (as mentioned) are essential for the type of analysis taken by the authors, they can take other authors/categories, but it’s necessary to define their approach.

Methodology
The methodology part has to be significantly improved. The authors do not
bring the clear information about taken research approach and chosen methodology.
Basically there is no clear theoretical base. Authors take qualitative approach, theymention the specific method (leveraged a voice-centred relational), but they do notshow their ontological and methodological basis. They clearly take some elements of discourse analysis. They also take some element from narrative approach. The taken methodology should be named clearly – is it “discourse analysis” – if yes, than whose approach do they use (which of the “discourse analysis” schools they use:
French one or Anglo-Saxon one). If they take communicative and linguistic method
of a narrative approach it should be stated directly with showing authors approach
(Lawrence, Riessman, Bruner).

Research Method
The first part (paragraph) is not about method, but about participants – it should beseparated or the title should be changed into e.g. “Participants and method” or Research process and method”

4
- The focus group took 35 minutes, it’s extremely short (less than 10 minutes per person without organisational matters and additional questions), while thinking about narrative approach it’s doubtful to gather proper data within so short time.
- When focus group was conducted?

- Lines 204-205: “Aside from reading a brief introduction to the study...” – what was
it about, it’s important to know how the focus group was introduced and started. To
know the topics signalized

Voice-Centred Relational Method
How the data was analysed – I mean in technical terms – what sort of data analysis
technique was used by authors (thematic analysis?). How did they emerge the
categories and themes – context analysis of words, phrases, sentences?
About “I” and “me” analysis – shouldn’t it also take into account “we’s” – I mean
using it to highlight the belonging to the group (while talking about “us”, “we” =
transgender man)

Summary of Participants –Background Information
There are some ethical considerations at this part about the research sample. It
was focus group interview conducted with three students (Alex, Naomi, and
Kristen)”. Who conducted the interviews? What about ethical consideration – were
they informed about purpose of the interview, way of using the data (what was the
instruction). What about anonymousness – we have 3 names, and a lot of data about
origins - are those nicknames and is all the data without misleading information?
How old were the participants –with the description of their stories it’s
difficult to get this information.

Stories from the field- -regulation and resistance
I suggest to make the title with “Research results: story from the field- regulation and
resistance” or similar
- Lines 250-259 – I do not understand placing this content at this section – It would fit
better to the introduction or for the discussion/conclusion

The paper has been edited and redrafted in response to the above highlighted comments

- Stories – it would be appreciated to get detail about the stories language – was it
written according to the original pronunciation, grammar ect. (as this story was
rewritten and reconstructed by researcher, and then edited by participant – if I
understood correctly – than the question is how close it is to the original statements

from interview. The stories seem not structured – it would be appreciated if some
blocks/topics would emerge from them (and be signalized as separate paragraphs).

This has been addressed

Discussion -Beyond the straight and narrow

This part of the article is not readable. While first two paragraph are clearly
distinguished, the remaining content is not structured. There should be clear and
logical structure implemented. Currently it’s switching from one topic to another

with mixing data with other researchers on such a way, that it makes difficulty to
understand which parts are authors, which parts are students, and which were
delivered from other researchers. It make this part vogue and difficult to transfer into
conclusions. The structure should be introduced according to the topics. There
should be clear distinction between participants thoughts, other students thought
and other research results and concepts (as those are also introduced and make it
more difficult to know it the terms were used by participants or by other authors. It
is also sometimes difficult to follow the coherence between content of participants
narratives and the categories used for their interpretation – the link is not obvious
(e.g. line 448-452: after description of difficulties with transition at their school “they
agreed that a faith-based schooling environment was not able to support transgender
people, due to systematic cisnormativity. Their fears, whether contemplating their
own secondary schooling contexts, or their existence in a contemporary context, are
unfortunately substantiated by a recent study in which 60.2% of LGBTIQ2SA+
participants, aged 14 to 21, had felt unsafe or uncomfortable at school or university
[68].” Did the students use term “cisnormativity”. We do not know why those
people in last research felt unsafe or uncomfortable, and that is probably not due to
taking decision about not coming-out
Peer and Teacher Intervention and Australia-a“Pretty Good Place to Be Trans”?
The beforehand comments apply also to those two parts (clear narrative, necessary
paragraphs following the logic of narrative and categories of analysis.

This have been amended and redrafted

Conclusion
It could be broaden and deepen – the elements of direct examples of participants
should be not addressed (no details about students), it should be pure conclussion.
The conclusion should be broaden with limitation of the research – number of
respondents, all respondents were from religious schools. In conclusion it would be expected to mention further research plan, as the authors showed rather initial analysis, so what are the further plans (initial analysis shows that its intriguing starting point for further research and exploration).

This have been amended  and  clarified as suggested

Many thanks for your thorough reading of our paper and for your detailed comments. We appreciate the generosity of time that you have spent assisting us to produce a revised paper that is clearer for the reader to follow

Round 2

Reviewer 3 Report

The article has been highly improved! The overall merit is high. The article still demands some significant improvements before publication. The main issue is the structure and description of the Methodology, which demands significant rewriting.

In the file I've attached, you will find detailed comments and suggestions. I believe that after their implementation, the article will be ready for publication. 

Author Response

Title and abstract The title and the abstract adjusted. Most of comments were taken into consideration. However, some issues were not resolved. The missing elements of the introduction: - there could be clear given clear definition of transgender/ cis gender people -now included . (Although authors bring some short contextual descriptions and broadly discuss the topic of “transgeneder” regulations and usage in Australian context and also bring some descriptions of this discourse in the  rhetoric, of Australian and Anglophone Media Discourse and Conclusions     I would still expect clear definition in Introduction- authors focus strongly on Australian context, which is well described (with some elements from other countries experience, e.g. British). There should be at least paragraph in introduction about the global changes in perception, legislation over the topic of transgender man (e.g. Istanbul Convention, changes in DSM classifications should be mentioned, with showing process of depenalization and overall switch into more inclusive global trends);now included I still consider it as important to add the global changes context some elements ofmedia discourse and global context are mentioned in the second part of the article.

This global context could be stressed strongly in Introduction it could be a 2 paragraphs about the global changes and challenges, as a lot have been done in the matter (especially on the level of cross-national legal regulations and medical approach) and it should be addressed  now included

- authors should deliver some theoretical background over the perspective of analysis. It’s about overall theoretical background and also about the detailed elements. now included

I would still expect a paragraph in Introduction concerning the clear description of their research methodology background it’s introduced in lines 65-69, but it should be still developedLines 52-64 they should be removed from this part and used in Methodology.Instead of this, there should be a brief information about research idea, purpose and methods now included and Lines 65-69. 52-64 have been addressed as suggested Section 2 and 3 are very well written   The methodology part has to be significantly improved. addressed as per the suggestions

  - Lines 184-195 (description of research approach - life history approach) - Lines 52-70 (description of method - focus groups, interviews)- Lines 159-176 (description of research procedure and participants recruitment) - Lines 195-199 (description of data analysis those lines has to be rewritten and brief information about VCR method as data analysis should be given really short, with comment, that afterwards it was used to reconstruct the narratives of interviewees - Lines 177-182 description of participants it can be left as a final part of the methodology section.   I couldn’t find information about: Who conducted the interviews? Ethicalconsideration how the interviewee were informed about purpose of the interview,addressed way of using the data (what was the instruction). What about anonymousness, How old were the participants Prior to the commencement of the focus group, the participants provided background 178 information regarding their educational histories- see Fig 1:There should be information about verbatim description of the table contentLine 179 Fig 1: Responses to preliminary questions asked of participants  

The figure could be made more readable (I offer a pattern which could be used: Suggested pattern used 5. Life History :Working with Narrative Stories from the fieldAfter taking some elements of section 5 to the Methodology part it would becomeslightly shorten and a bit of rephrasing will be necessary.Lines 618-633 : The limitation of the research could be a separate point it’s just for authors consideration addressed   Many thanks for your detailed and insightful comments which have been addressed. They have been very helpful in assisting us to revise and make the article stronger acknowledging how time poor many academics are your generosity of time and commentary is very much appreciated

Round 3

Reviewer 3 Report

The article is well improved. I suggest some slight changes – it’s just for your consideration:

1)      Point 5, “Method” is, in fact, part of Methodology, so line 230 could be deleted (and that part would be just a paragraph of part 4

2)      Point 6 “Participants” – a title like “Participants and data analysis strategy” or “Participants and research procedure” would fit better to the content of this part

3)      Line 271: 6.2. Voice-Centred Relational Method and analysis. – maybe “Data analysis strategy: Voice-Centred Relational Method”

4)      Line 301 – the stories of Alex, Kristen and Naomi should be a separate part of the article – they absolutely do not fit part 6.2.

I suggest adding subpart 6.3. with a title like, e.g.: “6.3. Research results: graduates stories” and give there a sentence of a brief introduction

Congratulations!

Author Response

Thank you once again for your feedback. Addressed comments 1-4